# Engineered Vasculature for Cancer Research and Regenerative Medicine

**DOI:** 10.3390/mi14050978

**Published:** 2023-04-29

**Authors:** Huu Tuan Nguyen, Arne Peirsman, Zuzana Tirpakova, Kalpana Mandal, Florian Vanlauwe, Surjendu Maity, Satoru Kawakita, Danial Khorsandi, Rondinelli Herculano, Christian Umemura, Can Yilgor, Remy Bell, Adrian Hanson, Shaopei Li, Himansu Sekhar Nanda, Yangzhi Zhu, Alireza Hassani Najafabadi, Vadim Jucaud, Natan Barros, Mehmet Remzi Dokmeci, Ali Khademhosseini

**Affiliations:** 1Terasaki Institute for Biomedical Innovation, Los Angeles, CA 90064, USA; 2Plastic, Reconstructive and Aesthetic Surgery, Ghent University Hospital, 9000 Ghent, Belgium; 3Department of Biology and Physiology, University of Veterinary Medicine and Pharmacy in Kosice, Komenskeho 73, 04181 Kosice, Slovakia; 4Bioengineering & Biomaterials Group, School of Pharmaceutical Sciences, São Paulo State University (UNESP), Araraquara 14800-903, SP, Brazil; 5Biomedical Engineering and Technology Laboratory, PDPM—Indian Institute of Information Technology Design Manufacturing, Jabalpur 482005, Madhya Pradesh, India

**Keywords:** microfluidics, engineered vasculature, organ-on-a-chip, cancer, regenerative medicine

## Abstract

Engineered human tissues created by three-dimensional cell culture of human cells in a hydrogel are becoming emerging model systems for cancer drug discovery and regenerative medicine. Complex functional engineered tissues can also assist in the regeneration, repair, or replacement of human tissues. However, one of the main hurdles for tissue engineering, three-dimensional cell culture, and regenerative medicine is the capability of delivering nutrients and oxygen to cells through the vasculatures. Several studies have investigated different strategies to create a functional vascular system in engineered tissues and organ-on-a-chips. Engineered vasculatures have been used for the studies of angiogenesis, vasculogenesis, as well as drug and cell transports across the endothelium. Moreover, vascular engineering allows the creation of large functional vascular conduits for regenerative medicine purposes. However, there are still many challenges in the creation of vascularized tissue constructs and their biological applications. This review will summarize the latest efforts to create vasculatures and vascularized tissues for cancer research and regenerative medicine.

## 1. Introduction to Vascular Engineering: Strategies and Techniques

Tissue engineering has made major impacts on cancer research and regenerative medicine through the development of new physiological and disease models [1], and therapeutic scaffolds that perform wound repair and drug delivery [2,3]. Still, one of the major hurdles in tissue engineering is the vascularization of large (>1 cm^3^) tissue constructs that deliver oxygen and nutrients to cells in the tissue [4]. Vascularized engineered tissues, inspired by the structure and function of the natural vascular system that facilitates nutrients and oxygen exchange between cells and blood vessels, are created by combining organ-specific cells and vasculature cells, scaffolds, and biologically active molecules to form functional tissues [5,6]. They have been used to study vascular pathophysiology, vasculature–organ interaction, and drug and cell trans-endothelium trafficking. In particular, they have been applied in cancer research to study angiogenesis [7], vascular remodeling [8], and metastasis [9,10]. Moreover, engineered vascular conduits that are usually large vessels (>1 cm^3^) have been used for regenerative purposes, replacing large dysfunctional arteria [11,12]. In the first section, an overview of vascular biology, as well as materials and methods for the creation of vascularized tissues, are reviewed, and in the second and third sections, we will focus on their recent applications in cancer research and regenerative medicine.

### 1.1. General Overview of Vascular Biology

The vascular system of the human body comprises arteries and arterioles constituting the arterial system, veins and venules constituting the venous system, and small capillaries that transport blood cells bringing oxygen and nutrients to the cells and carrying away carbon dioxide, lymph, and other waste [13]. High-pressure blood from the heart passes by the large arteries first, then small arteries and arterioles, before entering the capillaries, and then back to the heart by passing through the venules and veins [14]. Arterioles, capillaries, and venules form the multi-level vascular bed, which delivers nutrients and oxygen through the highly permeable vascular wall [15]. Lymph capillaries are the initial lymphatic vessels responsible for absorbing lymph, apoptotic cells, cellular debris, and circulating immune cells into the lymphatic system [16]. Endothelial cells (ECs) are common to all vessels and form the inner tunica intima layer that lines the vascular lumen. The middle layer, or tunica media, consists of circumferentially oriented smooth muscle cells (SMCs), and in large elastic arteries, multiple circular smooth muscle layers alternate with rings of elastic lamellae. Most blood vessels consist of three histologically distinct regions, each containing variable amounts of SMCs and elastin [13]. The cellular and extracellular matrix (ECM) composition of arteries, veins, and capillaries are different due to their function. Arterioles have fewer smooth muscle layers, and capillaries are covered by a discontinuous coat of pericytes (PCs) instead of SMCs. In contrast, veins have less smooth muscle and thinner walls than arteries and are more elastic [17].

The formation of new blood vessels, both in the embryo and postnatally, involves three major processes: (1) vasculogenesis, which is the formation of the first primitive vascular plexus in an embryo or postnatally de novo, (2) angiogenesis, consisting of the sprouting of new vessels from pre-existing ones, such as growing new capillaries out of postcapillary venules; and (3) arteriogenesis, defined as the rapid proliferation of pre-existing collateral vessels, induced by the change of blood flow during the obstruction [18,19,20]. Angiogenesis can be classified into two broad categories: sprouting angiogenesis and angiogenic remodeling [21]. After initial blood vessel formation, the vascular network is expanded, remodeled, and then undergoes maturation. Vascular tubes become covered by PCs on smaller capillaries or SMCs on larger vessels and ECM and become established functional blood vessels [22,23].

Angiogenesis was first observed in vitro by Folkman and Haudenschild, who observed the spontaneous organization of capillary ECs into capillary-like structures (CLS) and the formation of lumen [24]. Since these first angiogenesis assays, several achievements in engineering in vitro vasculatures have been done and reviewed previously [12]. In vitro vasculatures usually consist of three key elements: (1) human ECs, used to line the lumen of the bioengineered vascular structures; (2) human perivascular cells, used to support ECs function and/or provide perivascular stability to the networks; and (3), a scaffold that provides a physical space for the cells to interact and for the vascular network to develop, and the proangiogenic factors (VEGF, FGF and other various angiocrines) supplemented in the culture media or secreted by co-cultured stromal cells, such as fibroblasts [25].

The morphology and gene expression of the ECs is tissue-specific [26]. Endothelial cell phenotypes can differ not only between organs but they can also vary in the same organ between different areas of blood circulation, its neighboring ECs, and blood vessel subtypes [27]. This allows ECs to carry specific tissue functions [26]. For example, the brain endothelium and the BBB, respectively, consist of tight junctions ensuring low permeability and high transepithelial/transendothelial electrical resistance [28]. On the other hand, in order to provide a rapid molecular exchange, fenestrated endothelial layers can be found in kidney glomeruli and discontinuous endothelial layers in liver sinusoids [29].

The blood vessels’ differences in size, morphology, and function in different organs are long known, though distinct EC subpopulations are only recently being identified [29]. ECs express phenotypic heterogeneity depending on the location in the vascular tree [30]. Marcu and col. (2018) were able to point out different expression patterns of gene clusters involved in organ development and their functions. Data were collected mainly using isolated human fetal heart, lung, liver, and kidneys and their EC populations. The study has appointed that EC heterogeneity also translates into differences in their metabolic rates, with the heart ECs possessing the highest metabolic rate of the four aforementioned EC types [31]. Likewise, EC heterogeneity translates in the tumor environment. In a large study, lung tumor EC phenotypes of different species (human and mouse), patients, and in vivo/in vitro models were detailed by single-cell RNA sequencing (scRNA-seq) [32]. The scRNA-seq was performed on ECs of human/mouse and cultured lung tumor ECs, leading to the identification of previously unrecognized phenotypes and additional tip cell signatures.

One of the interesting tools used for the analysis of gene expression and mRNA translation in in vivo targeted cells is ribosome tagging (RiboTag). Targeted polyribosomes are labeled by hemagglutinin A and then isolated by immunoprecipitation, followed by qRT-PCR or RNA-seq. For example, Jambusarie and col. (2020) used RiboTag to isolate tissue-specific mRNA to analyze organ-specific ECs from collected brain, heart, and lung tissues to study the translatome patterns of gene clusters during homeostasis [33]. Furthermore, by exposing mice to endotoxin lipopolysaccharide, the study was also able to show tissue-specific gene expression related to vascular barrier function, metabolism, and substrate-specific transport [26]. Another study, presented by Cleuren and col. (2019) employed endothelial-specific translating ribosome affinity purification (EC-TRAP) and high-throughput RNA sequencing analysis. They demonstrated methods of in vivo snapshot and expression profiling and identified 82 shared genes among five vascular beds as well as pan-EC markers [14]. Understanding the differences and roles of individual EC populations can help improve novel treatment options and should also be considered when developing in vitro models [29,30].

### 1.2. Engineering 3-Dimensional In Vitro Vasculatures

Monolayers have been widely used in research over the years; however, the disadvantages of testing on (two-dimensional) 2D cultures have become more evident. The absence of hierarchical structure, nutrition gradients, limited cell–cell interactions, and cell organization in 2D cultures often result in higher sensitivity of the cells to drug treatments. Furthermore, the cells in (three-dimensional) 3D models behave differently than in 2D due to retained ECM signals. Therefore, 3D models often represent more physiologically relevant, in vivo-like microenvironments [34,35,36]. The recent progress in creating 3D cultures has brought several manufacturing methods, such as multicellular spheroids, organoids, scaffolds, 3D hydrogel scaffolds, organs-on-chips, and 3D bioprinting [37]. The efforts and progress in creating 3D cultures that would recapitulate the human organs are limited by missing vascularization within the structures. Due to a lack of proper nutrition and gas exchange in the thicker constructs, the cells might experience necrosis. At the same time, vasculature plays a pivotal role in many diseases, such as cancer metastasis, tumor angiogenesis, or atherosclerosis. The development of 3D models for drug testing, toxicology assays, in vitro imitation of pathological states, and progress in their applications for regenerative medicine are currently dependent on vascularization strategies [34].

#### 1.2.1. Vasculatures in Organoids and 3D Cultures in Hydrogel

Organoids are 3D tissue structures developed by using stem cells or organ progenitors cultivated under in vitro conditions, recapitulating key aspects of in vivo organs [37,38]. The derived cells are treated with growth factors and morphogens combined with appropriate matrices needed for distinctive developmental stages of organoids. Developed organoids are then cultivated using media with defined compositions, maintaining the intentional structure and function [38]. When engineering vascularized organoids, the matter of EC tissue-specificity should be considered as ECs characteristics proved to vary based on the represented tissue [28]. In general, the formation of vascularized organoids can be approached by in vitro or in vivo strategies. Obtaining vascularized organoids through in vitro strategies commonly refers to self-organizing and templating methods. Self-organizing methods imply the self-assembly of co-culture consisting of supporting cells and ECs [39]. For example, embryonic stem cells were used to produce self-organized human blood vessel organoids and brain organoids, both induced from human embryonic stem cells [40]. The brain organoids and blood vessel organoids were generated separately and then fused together by placing two blood vessel organoids on two sides of the brain organoid. Interestingly, the fusion has led to the display of not only a blood–brain barrier-like structure but also the presence of microglial cells, creating a model of interactions between neuronal, vasculature, and microglial components.

New approaches to in vitro formation of vascularized organoids are constantly being developed. Recently, the induction of stem cell differentiation can be initiated by controlling the media composition or by regulating the intracellular state through the overexpression of various transcription factors. The transcription factors-based protocols, however, usually apply cues favoring differentiation into a single cell line. This restrains the simultaneous differentiation of human-induced polypotent stem cells (hiPSCs) into multiple cell lines, hierarchically patterned structures, and, therefore, the generation of more complex multicellular tissues. To overcome this problem, Skylar-Scott et al. (2022) used an orthogonally induced co-differentiation method to produce ECs and neurons from hiPSCs by the overexpression of transcriptor factors, all in a one-pot system followed by organoid generation [41].

Finally, the alternative to in vitro strategies is vascularization in vivo by transplanting organoids formed in vitro into a host [39]. Recently, Kim and col. (2022) have produced human kidney organoids using decellularized ECM (dECM) hydrogel derived from porcine kidneys and human iPSCs. 10–20 dECM-based kidney organoids were transplanted into the renal subcapsular space of the immunodeficient male NOD/SCID host mouse. 14 days after transplantation, robust angiogenesis sprouting from host blood vessels was observed and enhanced glomerular-like structures’ maturation in kidney organoids was shown [42].

#### 1.2.2. Conventional, Additive Manufacturing and Bioprinting of Vasculature Tubes

To fabricate a basic construct resembling a blood vessel, one could create a tubular structure within a biomaterial and subsequently seed the channel with ECs. This section will discuss the different methods for manufacturing tubular structures with a diameter of more than 100 microns.

The first works related to vascular graft manufacturing, creating tubular structures by molding biomaterials and cells into cylindrical shapes using conventional manufactured sacrificial templates (CMST) [43,44]. In 1986, Weinberg and Bell were the first authors to report a tissue-engineered blood vessel. They cast a mixture of culture medium, collagen, and SMCs in an annular mold [45]. In 1998, L’Heureux et al. introduced the rolling sheet method. In this method, cells are cultivated in culture falcons with ascorbic acid supplementation for four weeks to form a cell-made biomaterial sheet. This sheet is wrapped around a mandrel and cultivated under flow culture for 8 weeks, transforming the cell sheet into a robust tissue-engineered blood vessel [11].

However, the variety of producible shapes is quite limited for the CMST, e.g., creating a complex branched vascular network is challenging [46]. To solve this shortcoming, additive manufacturing methods, such as 3D Printing, Selective Laser Sintering (SLS), and Stereolithography (SLA), can be used to create more complex additive manufactured sacrificial templates (AMST) [44,47,48]. For example, carbohydrate glass can be 3D printed or mixtures of isomalt and cornstarch can be laser melted by SLS to form a sacrificial template based on a 3D computer-aided design (CAD). This template is subsequently embedded in a hydrogel. After solidifying the hydrogel, the sacrificial template is dissolved, and a tubular network is left on the inside of the hydrogel [44,47]. In stereolithography, a CAD design directs light on a vat of photopolymerizable liquid hydrogel in a layer-by-layer fashion. The part of the biomaterial that is not solidified functions as the sacrificial template and is leached out to create the channel structure within the hydrogel [48]. Even so, CMST and AMST do not allow the exact positioning of multiple cell types, biomaterials, and signaling molecules, which is necessary to create organ structures with a complex 3D organization of multiple cell types. A 3D bioprinter equipped with multiple printing heads, however, can create such complex constructs. Each printing head can be loaded with a specific bio-ink, which can be subsequently deposited in a specific 3D location on the printing platform. A bio-ink is a printable combination of a biomaterial, cells and/or biomolecules, i.e., proteins and growth factors [49].

The 3D bioprinting methods to fabricate vasculature tubes can be divided into two main groups: direct and indirect. In indirect bioprinting, one printing head prints a vascular pattern with a sacrificial ink that functions as the sacrificial template. Another printing head subsequently prints vascular bio-ink around the sacrificial ink [50]. Materials such as (non-functionalized) gelatin and pluronic F-127 have already been utilized as sacrificial inks. After printing, these sacrificial inks are dissolved to leave a tubular structure behind [50,51]. By contrast, in direct bioprinting, the vascular ink and the sacrificial ink are printed simultaneously to “directly” print the desired tubular shape using a coaxial printing head [52]. When comparing the current state of 3D bioprinting to AMST, one could state some drawbacks about the 3D bioprinting method; 3D bioprinting is a slower process than AMST. Cell viability becomes a concern because bioprinting an organ takes several hours [47]. The typical materials used for cell encapsulation and 3D bioprinting are also vulnerable to deformation. Overhanging structures are more likely to collapse under their weight. Their viscosity and surface tension also limit the precise extrusion of smaller volumes. This makes the current 3D bioprinting methods less attractive to recreate predefined vascular shapes and less suitable for long-term culture under deforming forces, such as fluid flow [44,48]. However, a new promising strategy called multimaterial SLA could be the solution to this problem. This method enables SLA to print multimaterial constructs by switching between vats of different photo-polymerizable liquid bioinks with rinsing in between to avoid mixing the different bioinks. In this way, although limited to photo-crosslinkable materials, multimaterial structures can be patterned without the material constraints as seen for bioprinting, and overhanging structures like vasculatures can be more easily printed [53]. We summarize here all the above-mentioned fabrication methods in Table 1.

#### 1.2.3. Vasculature-on-a-Chip

Organ-on-a-chip (OOC) is an emerging technology for modeling human physiology in a miniature system [76]. Vasculature-on-a-chip, in particular, allows recapitulation of certain aspects of in vivo vasculature, including the EC polarity, basement membrane, as well as fluid flow, and molecular or cellular transportation across the endothelium [77]. This section will focus on the methods for creating vasculatures and fluid flows within a microfluidic system, as well as sensing methods for evaluating vasculature’s physical and biochemical markers.

##### Creation of Vasculatures in Microfluidic Devices

Microfluidic devices allow the control of several physical and biochemical cues, such as ECM stiffness, interstitial and perfusion flows, and molecular gradients, while offering the possibility to image EC behavior in real-time and high resolution. Therefore, researchers could create vasculatures of different sizes ranging from micro (typically ∅10 µm diameter) or macro-(typically more than ∅100µm) vasculatures [4]. Compared to AMST, CMST or bioprinting, microfluidic-based vasculatures are encapsulated inside a channel, therefore, can be interfaced with external pumps or reservoirs, allowing controllable vasculature perfusion [78,79].

There are two main methods to create vasculatures within a microfluidic device: top-down and bottom-up [80]. The top-down approach consists of populating ECs inside a pre-designed hollow structure within a gel or on a surface, while the bottom-up approach uses cellular or extracellular stimuli to promote the self-assembly of ECs within a gel. Common top-down methods to create hollow structures within a gel laden or within a microfluidic chip are layer-by-layer fabrication, laser-degradation, 3D printing, and rod/needle removal [81]. The latter, pioneered by the Tien group, consists of removing a needle or Polydimethylsiloxane (PDMS) rod from a cell laden within a microfluidic chip, creating a microfluidic channel followed by the seeding of ECs inside. Top-down vascularization approaches conducted by tissue engineering methods such as hydrogel molding by needles, sacrificial molding, or bioprinting are considered templating methods [82]. Sacrificial writing into a functional tissue biomanufacturing method that can be used for biomanufacturing organ-specific tissues with embedded vascular channels was introduced [39,83]. Skylar-Scott et al. (2019) introduced sacrificial writing into a functional tissue biomanufacturing method that can be used for biomanufacturing organ-specific tissues with embedded vascular channels [83]. Organ building blocks (OBBs) represented by organoids, multicellular spheroids, or embryoid bodies were placed into engineered ECM and compacted by centrifugation, resulting in a living OBB matrix. Perfusable vascular channels were then created via embedded 3D bioprinting using sacrificial gelatin ink. When exposing the construct to 37 °C, ECM underwent gelation while the sacrificial ink was removed, creating a network of tubular channels. Endothelial-lined channels can then be obtained by perfusion of human umbilical vein endothelial cells (HUVECs) suspension through the embedded vascular network [83]. Another recent example of the top-down approach is the study by Humayun et al. [84]. In this study, an intestinal OOC was created by seeding primary intestinal epithelial cells and ECs within hollow structures created by removing PDMS rods. Similarly, Lugo-Cintrón et al. created a microfluidic device to understand the effect of ECM density on lymphatic vessels [85]. The device comprised two PDMS layers, a milled microchamber, and a rod between the media ports. After polymerizing a hydrogel in the main chamber, the rod was removed, leaving empty space for seeding human lymphatic ECs. Alternatively, bottom-up approaches perform endothelial seeding in the ECM, allowing them to create interconnected microvasculatures via self-assembly. ECs and stromal cells such as fibroblasts, when seeded inside fibrin, could become perfusable microvasculatures. Discovered firstly by the Jeon group, this technique is widely adopted for creating in vitro microvascular beds [86]. An example of this approach is a study by Hachey et al., which developed a 3D tumor model on a chip to simulate a human tumor microenvironment [87]. Other examples and applications of bottom-up methods will be presented extensively in the following sections (vide infra). Self-assembled microvasculatures have small diameters, typically in the range of ∅10µm, compared to vasculatures created by top-down approaches, which yield typical diameters of ∅100 µm. However, the morphology and structures of vasculatures created by using bottom-up approaches are not easily controlled, unlike those created by a top-down approach. Therefore, large vessels, mimicking arterioles or venules, can be created by using pre-defined structures, and vascular beds, which are smaller in size, are better recapitulated using a self-assembly approach. Combining these two approaches can generate multi-scale vasculatures, representing the complexity of in vivo vasculatures [88,89]. We have summarized the methods for creating in vitro vasculature in Figure 1.

In addition to fibrin, which is known to be angiogenic, other natural ECMs have been used to culture ECs, such as collagen and fibronectin [82]. Synthetic hydrogels are becoming a viable alternative to natural ECM thanks to their reproducibility and tunable mechanical and biochemical properties, as reviewed previously [90].

Various ECs sources are used for engineering vasculatures, with HUVECs being commonly used along with other primary human or animal ECs. Stromal cells such as SMC, fibroblasts, mesenchymal stem cells, pericytes, and astrocytes are often used to create different models of macro or microvasculatures. The appropriate combination of ECs and stromal cells is discussed elsewhere [91].

##### Methods for Generating Fluid Flows and Performing Physical and Biochemical Sensing

One of the main advantages of microfluidic systems is the ability to create fluid flows to study the effect of flows on living organisms, cells, or tissues in real time. Commonly, researchers performed flow experiments using microfluidic-based vasculatures to investigate drug transports or ECs’ responses to various molecules under physiological flows. Wang et al. reported a device to model the leaky vasculature of tumor tissues and study its effects on the perfusion of therapeutic-sized nanoparticles (NPs) [92]. The chip was composed of two microchannels separated by a porous PDMS membrane. The top channel modeled the microvasculature of tumor tissues and was seeded with HUVECs, whilst the bottom channel consisted of a large chamber surrounded by rectangular pillars housing a mixture of tumor spheroids composed of human ovarian cancer cells (SKOV3) and collagen to model the ECM of tumor tissue (Figure 2A). A perfusion rate of 0.64 μL/min provided optimal shear stress to promote the elongation of ECs in the direction of flow and the development of intercellular junctions. Impaired vascular permeability was mimicked using tumor necrosis factor-α (TNF-α), verified via VE-cadherin imaging and 70 kDa dextran assay. Next, the perfusion of liposome NPs and poly(ethylene glycol)/poly(lactide-co-glycolic acid) nanoparticles (PEG-PLGA NPs) through the endothelial and ECM barriers was studied, as they account for most clinical nanotechnology-based therapeutics. The movement of these NPs through the vasculature and tumor ECM was similar to that of anti-cancer therapeutics in vivo, which are slowed down by the dense tumor tissue [93,94].

In addition to shear flow, interstitial flow (IF) also plays a crucial role in tissue homeostasis. Kim et al. investigated the effects of the presence and direction of IF on angiogenic sprouting [95]. Using a PDMS model, Kim found that pro-angiogenic factors synergize with IF moving against the direction of sprouting, encouraging the development and sustained growth of angiogenic sprouting in a HUVEC and stromal fibroblast co-culture. When the flow was reversed, initiation of angiogenic sprouting was suppressed and growing sprouts were regressed. Furthermore, the effects of VEGFR pathway-mediated anti-angiogenic factors were found to be more effective in preventing initiation and further growth of angiogenic sprouting in the presence of IF, further revealing the role of IF in the movement and uptake of biochemical signals.

PDMS is a suitable material for engineering vasculatures in microfluidic devices; however, it also suffers from major drawbacks such as non-specific molecular adsorptions and lack of degradability. Therefore, new materials are being explored for engineering vasculatures. Zhang et al., as shown in Figure 2B, employed a biodegradable elastomer poly(octamethylene maleate (anhydride) citrate) (POMaC) device capable of modeling complex vasculature in a variety of different ECM [96]. Thin, pre-patterned POMaC sheets were stacked layer-by-layer under UV light, creating a crosslinked scaffold containing 100 μm by 100 μm 3D microchannels with a wall thickness of 25–50 μm. The scaffold proved to be thin and flexible yet durable. To promote biomolecular exchange, cell migration, and porosity, 10 and 20 μm channels were patterned along the scaffold wall, together with nanopores throughout the polymer, to allow for oxygen and nutrient exchange. Flow was driven by a pressure–head difference between the inlet and outlet wells. The inside of the scaffold was seeded with HUVECs. The confluence of the endothelial monolayer was confirmed using CD31 staining. The 70 kDA FITC-dextran assay revealed a membrane permeability similar to mammalian capillaries. The reader is encouraged to read Polacheck and colleagues (2013) for a more comprehensive review of the development of microfluidic systems to model mechanobiology [97].

The incorporation of sensors in 3D and microfluidic cell culture is an emerging feature of microfluidics devices that allows for further real-time study of these barrier models [98]. Zhang et al. created an OOC platform for automated monitoring and control of organoids using electrochemical and physical sensors and microscopes (Figure 2C) [99]. Built-in microchannels and pneumatic valves made the device programmable and allowed for continued fluid flow (Fluid path scheme shown in Figure 2D). Regeneratable electrochemical sensors utilized the electron transfer kinetics between redox electrodes and antibody–antigen binding systems to monitor organoid-secreted biomarkers. The pH sensor detected the optical differences in the color of phenol red at different pH levels of the culture media, whilst the oxygen sensor monitored the oxygen-sensitive fluorescence of ruthenium dye in the media. Furthermore, mini microscopes positioned at the bottom of the bioreactors allowed for real-time monitoring of organoid morphology. It is important to note that the above electrochemical biosensor can be used to monitor several biomarkers at once and can be applied to monitor almost any biomarker. Therefore, in the future, vascularized tissue models could be monitored using an electrochemical biosensor for real-time assessment of vasculature biomarkers. A further review of the integration of OOC technology for vasculature modeling is discussed by Lee et al. and O’Connor et al. [89,100].

## 2. Application of Engineered Vasculatures in Cancer Research and Drug Delivery

Successful cancer treatment entails novel drugs and delivery approaches [101]. Drug development requires various screening models, including cell culture on a plate and animal models [102,103]. However, cell culture on a plate is limited in replicating the in vivo microenvironment and can only be used for early toxicity testing [104,105]. Animal models can replicate in vivo drug responses but are limited in their ability to predict drug side effects and effectiveness in humans [76,106].

The tumor microenvironment (TME) promotes tumor growth and tends to be highly inflamed, characterized by the presence of various immune cells and the growth of blood vessels [107]. Moreover, the function of blood and lymphatic vessels in the TME is aberrant, as they are more tortuous, disorganized, and non-functional and contribute to tumor immunosuppression and resistance to treatment. Therefore, one strategy to treat tumors is combining standard therapy and anti-angiogenesis agents to normalize tumor vasculature transiently, enhancing the efficacy of anticancer drugs delivered during the normalization window [108]. As biomimetic in vitro systems can create a bond between 2D in vitro and animal models by imitating the 3D design of in vivo tissues, vascularized tumor models are engineered to test anticancer drugs [109]. A common practice is to culture vascular ECs with tumor spheroids to create a network of blood vessels [110]. These vascularized tumor models are employed to study (*i*) the delivery of nutrients and oxygen to the tumor, (*ii*) the efficacy of drug delivery systems, and (*iii*) tumor development and progression. Here, we summarize recent engineered vasculature models to study the mechanisms underlying tumor development and metastasis, as well as the transport of anticancer drugs and immune cells.

### 2.1. Angiogenesis and Tumor–Vasculature Interactions

Various in vitro, ex vivo, and in vivo angiogenesis models have emerged that can recapitulate the in vivo tumor microenvironment for understanding the disease pathogenesis, tumor–vasculature interplay, molecular interaction between tumor–stromal and vasculature, angiogenesis, or drug development [7]. Recently, many 3D models vascularized tumor models have emerged to improve efficient drug delivery and therapeutic responses [111,112]. These models are known as microphysiological models that can capture branching morphogenesis, tumor heterogeneity, and vasculature [7]. Seo et al. developed a 3D blood–brain barrier (BBB) composed of brain ECs, pericytes, and astrocytes and co-culture it with a glioblastoma spheroid (T98G). Tumor cells show more aggressive behavior when co-cultured with BBB, as quantified by angiogenesis and vessel formation (Figure 3A–C) [113]. Organ-on-a-chip models for BBB research were reviewed elsewhere [114]. Truong et al. engineered a 3D organotypic microfluidic platform to create microvasculature using patient-derived glioblastoma stem cells (GSCs) cells to investigate the role of ECs on GSC invasion. They showed that microvasculature enhances cell migration, proliferation, and invasion phenotype [115]. A recent study reported a bottom-up approach to the tumor microenvironment to create a vascularized 3D breast cancer model to investigate angiogenesis and cancer invasion. This system demonstrates aspiration-assisted bioprinted tumor spheroids and studied cancer endothelial interaction collectively [116]. Some recent studies established that creating vasculature in tumors does not necessarily originate from EC cells [117]. However, most of the genetic alteration is associated with the angiogenetic phenotype. Dikici et al. developed an alternative synthetic biodegradable vascular network using poly-3-hydroxybutyrate-co-3-hydroxyvalerate (PHBV) to study angiogenesis. They used 3D printing to model human skin to investigate the importance of physiological relevance while studying endothelialization of the engineered skin tissue construct [118].

In vitro microfluidic chips have been designed to understand the role of hypoxia in promoting breast cancer metastasis. A microvascular network model was created in the microfluidic chamber, and HIF-1α knockdown breast cancer cell extravasation at different oxygen tension was studied by creating different cancer stages with MCF10A, MCF-7, and MDA-MB-231 [119]. In tumors, hypoxia facilitates angiogenesis by regulating growth factors such as vascular endothelial growth factor (VEGF) [120]. The hypoxia adaption mechanism is a well-known molecular pathway-driven vascularization mechanism. Ando et al. bioengineered a vascularized hypoxic tumor model to study angiogenesis using a microfluidics-based platform, using ECs in collagen gel that generated lumen around the 3D cancer tissue construct. This model created heterogeneity in the tumor by generating oxygen and metabolite gradient. The interaction of a tumor with vasculature was studied, which is the prominent feature of the solid tumor [121]. In addition to long-term culture possibility, high-resolution image analysis for vasculature formation has been demonstrated.

### 2.2. Drug Efficacy and Toxicity Evaluation

The tumor microenvironment plays a crucial role in therapeutic responsiveness. In vitro models that faithfully represent the drug administration process are necessary to improve drug delivery and efficacy. Moreover, for proper tissue functioning, a perfusion-based model would be required to include the effect of physical, chemical, and biological parameters. In pursuit of this, tremendous effort has been made to realize the role of the vasculature in the 3D microphysiological model that includes complex human biology in both healthy and pathological scenarios.

Hachey et al. developed a vascularized micro-tumor (VMT) model with a microfluidics platform using colorectal cancer for real-time drug response and tumor–stromal interaction study. They studied gene expression and tumor heterogeneity and showed that VMT treatment responses are closer to clinical pathological results when compared with 3D spheroid models or 2D models [87].

Furthermore, vasculature is created in liver tissue using agarose and GelMA to study the effect of the endothelial barrier during drug treatment [122]. In this work, the authors engineered a central vessel using agarose fiber to create a hollow capillary within a GelMA scaffold using the 3D printing method in a microfluidic device (Figure 3D,E). HUVECs were used in the microchannel to create an endothelial barrier, and the liver tissue construct was developed by encapsulating HepG2 and C3A liver cancer cell lines within GelMA (Figure 3F,G). They studied the toxicity effect of APAP, which has an effect on the endothelial barrier and is one of the major causes of liver failure. They perfused APAP for 48 h through the microchannel and observed significant damage in cell viability. Media with APAP perfusion damaged the HUVEC layer integrity, whereas the layer remained intact in the control (Figure 3H,I).

Similarly, isolated proximal epithelial tubules resuspended in hyaluronic acid hydrogel were used to emulate the vasculature of kidney tissue to study the toxicity of gold nanoparticles [123]. This study found that gold nanoparticles were unsuitable for use in these cultures and indicators for toxicity were observed, such as cytokines, Kim-1, and cell viability, when compared side by side to immortalized cell lines. On a similar note, Yang et al. resuspended 2D cell culture systems into PluronicF127 acrylamide–bisacrylamide hydrogels containing microsomes to help predict drug toxicity [124]. The authors tested the toxicity of the CTX drug on their hydrogel model with MCF-7 cells and found that higher CTX concentrations resulted in higher early and late apoptosis compared to their 2D counterparts.

### 2.3. Metastasis and Immune Cell-Vasculature Interactions

Metastasis is a deadly landmark in cancer and a multi-step process consisting of (1) the infiltration of tumor cells into the tissue adjacent to the primary tumor, (2) the intravasation of tumor cells into blood and lymphatic vessels, (3) survival in the circulatory system, (4) extravasation, and (5) colonization into distant sites [125,126,127]. Blood and lymphatic vessels play a crucial role in tumor metastasis [128,129,130]. On the other hand, vasculature is also an essential factor in therapeutic approaches, participating in the transport of anticancer drugs and immune cells [131,132,133,134].

A recent work by Kim et al. [8] described approaches for directing angiogenesis in lung cancer spheroids within microfluidic channels. Angiogenic sprouting from blood vessels was stimulated by adjusting the flow settings and the presence of lung fibroblasts. The microfluidic system included a collagen-type I/fibrin-based hydrogel where three parallel microchannels were formed. In addition, a PDMS-based portion of the microfluidic system was used to create six reservoirs (Figure 4A). HUVECs were seeded inside one microchannel, forming a lumen and maintaining their tubular shape within the 3D system (Figure 4B). The ECs express vascular endothelial (VE)-cadherin at the cellular junctions in capillaries that grew out of the main blood vessel (Figure 4C). After the vascular lumen formed, lung cancer spheroids were integrated into the microsystem through another channel (Figure 4D,E). In addition to employing the resulting vascularized lung cancer model in the efficacy testing of anticancer drugs, the movement of immune cells through the newly formed small blood vessels (sprouted capillaries) was examined (Figure 4F–H). The THP1 cell line, which is a type of precursor macrophage cells, was utilized as an example of immune cells. The research was conducted on tumor systems of the lungs that have and do not have blood vessel sprouting, and it was found that the THP1 cells (displayed in blue) only moved through the small blood vessels in the tumor system that has angiogenic vessels sprouting from the macro-lumen (Figure 4G,I).

In addition to microfluidics, 3D bioprinting is another powerful approach to designing and fabricating complex microphysiological systems [135]. Three-dimensional bioprinting allows for accurately integrating various arrangements of cells and ECM-like compounds [136]. Three-dimensional bioprinting offers new possibilities for creating vascularized tumor tissues [137]. In a recent work by Meng et al., 3D bioprinting was employed to position islets of cancer cells, mesenchymal cells, and vascular cells within a 3D microsystem (Figure 4J-i) [138]. Furthermore, extracellular chemical gradients were controlled by integrating stimuli-responsive capsules containing growth factors. The microchannels in the microfluidic system were injected with HUVECs (Figure 4J-ii,iii). Perfusion of a fluorescent compound that has viscosity close to human blood confirmed the vessel lumen (Figure 4J-iv). An islet of A549 cells was integrated into the vascularized system between microcapsule arrays. In contrast, fibroblasts were incorporated into the surrounding hydrogel to recapitulate the primary tumor (Figure 4J-v). One of the experiments evaluated the tumor cell migration guided by EGF-loaded microcapsules. It was found that the gradient created by the EGF microcapsules increased the proliferation, expansion, and directional movement of the tumor cells (Figure 4K). VEGF-loaded microcapsules were also integrated into the system to promote and direct the vascular network (Figure 4L). By triggering the VEGF release, ECs sprouted from the central vessel axis. Combining both EGF- and VEGF-loaded microcapsules within the vascularized tumor model produced metastatic dissemination of tumor cells with modulated sprouting angiogenesis (Figure 4M). The 3D bioprinted vascularized tumor system successfully models tumor invasion and angiogenesis by utilizing laser-triggered growth factor-releasing microcapsules.

Tumor cell extravasation and colonization of the distant tissue can be modeled using microfluidic systems and were reviewed elsewhere [139]. These devices commonly have a vasculature channel lined by ECs and perfused with cancer cells that perform trans-endothelial migration and mimic the extravasation process. These platforms are powerful in investigating the mechanism of extravasation and early colonization, especially the role of immune cells in this process. For example, in work by Boussommier-Calleja et al., the impact of monocytes on the movement of tumor cells out of blood vessels was studied using a 3D microfluidic system that simulates blood vessels [140]. The microfluidic system was composed of five microchannels connected to cell growth media reservoirs (Figure 4N-i). The microchannels enclose three chambers: hydrogel cell-laden in the central chamber and cell growth media in the left- and right-side chambers (Figure 4N-ii). The central chamber was filled with fibrin–HUVEC: fibroblast (6:1) cell-laden hydrogel. The ECs organized themselves spontaneously into a 3D network of blood vessels in less than seven days, connecting the side cell growth media microchannels. The cell growth media microchannels introduced monocytes and cancer cells into the vascularized system. The monocytes entrapped within the microvasculature lumen were shown to extravasate within 24 to 48 h (Figure 4O). The movement of cells out of the blood vessels (extravasation) occurred rapidly once it began. The main part of the cell passes through the wall of the blood vessels in less than 180 s (Figure 4P). After extravasation, monocytes populated the extravascular space and never re-entered a vessel (Figure 4Q,R). Utilizing this immune-competent vascularized system, the authors demonstrate that monocytes can directly reduce cancer cell extravasation in a non-contact-dependent manner.

A study by Kim et al. highlighted the development of a microfluidic platform that can be used to investigate the complex interactions between cancer cells, recruited immune cells, and the ECM in the pre-metastatic niche [141]. The platform allows for a more accurate representation of in vivo systems by incorporating ECs and ECM scaffolds. The research demonstrates that monocytes and macrophages play distinct roles in establishing pre-metastatic niches. Monocyte-derived MMP 9 facilitates cancer cell extravasation, and macrophages generate characteristic “microtracks” for cancer cell migration. These findings can potentially improve our understanding of the pre-metastatic niche and provide new insights into immunotherapies for cancer. The microfluidic platform developed in this study may also have practical applications in clinical workflows, allowing for more accurate and efficient testing of potential treatments for cancer patients.

Lymphatic vessels play a crucial role in the immune system and are also essential for maintaining fluid balance in the body. Recent studies have shown that lymphatic vessels also significantly impact tumor biology and cancer therapeutics [142]. Therefore, engineering in vitro lymphatic vasculature has become an area of great interest in biomedical research. Despite the high demand for such models, the development of in vitro lymphatic vessels is still in its early stages compared to blood vessel modeling. Researchers are exploring various methods to create functional lymphatic vessels in vitro, such as 3D bioprinting, microfluidic systems, and organ-on-a-chip technology. These efforts will pave the way for advanced studies of lymphatic biology, cancer metastasis, and the development of targeted therapies.

For instance, a microfluidic model was created to examine how breast cancer cells affect the gene expression of lymphatic ECs, which can lead to lymphatic dysfunction [143]. The study analyzed the impact of estrogen receptor-positive MCF7 and triple-negative MDA-MB-231 breast cancer cells on lymphatic vessels and found changes in gene expression related to vessel growth, permeability, metabolism, hypoxia, and apoptosis in lymphatic ECs. The results indicate that this type of microfluidic model can be useful for investigating interactions between breast tumors and lymphatic systems across different subtypes of breast cancer. In another study, a 3D in vitro model of human lymphatic vessels in a tumor immune microenvironment utilized an injection-molded plastic array culture platform to leverage high-throughput testing of lymphatic-related strategies for cancer therapeutics [144]. By optimizing the cellular composition and spontaneous capillary flow-driven patterning of 3D cellular hydrogel, the platform allowed for the robust and reproducible formation of self-organized lymphatic vessels, replicating the morphogenesis of lymphatic vessels in a cancer cell type-dependent manner. Additionally, the model’s robustness enabled high-content analysis to investigate the effect of anti-VEGFR3 drugs on different types of cancer cells or the existence of blood vessels. The development of in vitro lymphatic vessels is an exciting and rapidly growing field of research with enormous potential for advancing our understanding of lymphatic biology and cancer metastasis. These advances will undoubtedly pave the way for further research in this field, ultimately leading to the development of new targeted therapies and improved patient outcomes. 

Overall, it is critical to note that it is essential to strike a balance between complexity and simplicity when it comes to reconstructing an in vitro tumor immune microenvironment. While we can never fully replicate the intricacies of an in vivo immune system in a laboratory setting, simplifying the system can allow us to dissect specific mechanisms involved in tumor invasion and immune cell interactions. However, it is crucial not to oversimplify the system, as this can lead to inaccurate conclusions and hinder progress in the field. Ultimately, finding the right balance between complexity and simplicity is key to advancing our understanding of tumor immunity and developing effective therapies.

### 2.4. High-Throughput Screening

Assay throughput is an important factor to be considered in developing in vitro platforms, especially for drug screening applications, in which many molecules need to be tested efficiently and rapidly. Currently, OOC platforms are PDMS-based and possess low to medium throughput. This is partially due to the device fabrication process, which traditionally involves photolithography and soft lithography [145]. To this end, a scalable and robust fabrication method using 3D-printed molds has been demonstrated [146], eliminating the need for mask fabrication in cleanrooms. Another potential solution to improve throughput is to design, in place of single-unit PDMS-based microfluidic chips, plate-based platforms with each well housing a single chip unit. In fact, multi-well plate-based OOCs have previously been reported to demonstrate the potential of OOCs to scale up [147,148,149,150].

For the models of microvasculature specifically, several studies have used the two-lane or three-lane microfluidic plates commercialized by MIMETAS [151,152,153]. These OOC devices offer the capability to culture vascular ECs in 3D with perfusion, allowing them to self-organize and form microvascular networks embedded in ECMs. As an example, Jung et al. reported two perfusable lung microvasculature-on-a-chip models for the alveoli and small airway by tri-culturing human microvascular ECs, fibroblasts, and PCs [152]. The proposed plate-based platform provided the ability to test 64 chips at once for drug discovery and disease modeling applications. Apart from the commercially available options, Song et al. developed a high-throughput 3D model of tumor microvasculature with colorectal cancer cells (CRCs) and HUVECs [154]. The platform demonstrated its capability to assess the effects of natural killer cells on CRCs while enabling a 10-fold increase in the number of experiments that can be performed compared to PDMS-based microfluidic chips. High-throughput platforms for microvascular models have been used to screen for molecules with anti-angiogenic properties [155,156,157,158]. Duinen et al. developed a perfusable, 3D angiogenesis assay platform using human iPSC-derived ECs for drug screening [157]. Using one of the MIMETAS plate-based platforms, they were able to show successful VEGF-gradient-driven angiogenic sprouting. Moreover, assay performance was quantified by measuring a minimum signal window and Z-factor, which are suitable for drug screening purposes.

Improvement of the physiological relevance of OOCs by, for example, including additional cell types and linking multiple organ chips (i.e., multi-OOC) remains a major interest and focus of future OOC research; increasing the throughput of such platforms continues to be a challenge. Future endeavors to optimize this trade-off or achieve both will be essential for microvasculatures-on-chips to become widely adopted by the pharma industry and academic laboratories for high-throughput screening applications.

### 2.5. Patient-Specific Model

The idea of personalized medicine encompasses a system in which patients are treated, and clinical decisions are made based on inter-individual differences. It is well established that therapy response varies between patients largely due to genetic factors [159]. To this end, in vitro models constructed with patient-derived cells have contributed significantly to our understanding of patient-to-patient heterogeneity in disease progression and treatment response [160,161,162]. In this regard, the discovery of human induced pluripotent stem cells (hiPSCs) has provided an alternative source of cells to build patient-specific in vitro models [163]. In contrast with conventional biopsies to obtain cells from patients, hiPSCs involve a non-invasive approach to provide patient-derived cells in large quantities. Currently, several differentiation protocols are available and being explored to drive them into different cell lineages, including vascular ECs [164]. However, even with substantial advances made in the iPSC technology, it remains a challenge to achieve full maturity in differentiated cells [165].

In the context of microvasculature models, several studies have previously shown the use of iPSC or embryonic stem cell-derived cells as their cell sources [166,167,168,169,170]. Kurosawa et al. investigated the hiPSC-derived ECs as a potential source of cells to form 3D microvasculature within a microfluidic OOC device [167]. The iPSC-ECs showed several EC-specific phenotypes, such as the expression of key EC markers (e.g., von Willibrand factor, CD31, and endothelial nitric oxide synthase). Moreover, the inhibition of TGF-β signaling led to improved vascular formation by iPSC-ECs. More recently, Cuenca et al. co-cultured hiPSC-derived ECs and vascular smooth muscle cells (VSMCs) to realize a robust 3D vessel-on-chip model [168]. The hiPSC-ECs formed functional microvasculature in a fibrin hydrogel. For the characterization of hiPSC-CSMCs, automated image analysis was performed to measure intracellular Ca^2+^ release in the cells.

Patient-derived primary cells also have been used to develop ex vivo disease models with microvessels to assess drug response [171,172,173,174]. Patient-derived breast cancer organoids were previously cultured with a 3D microvasculature on a microfluidic device [172]. The authors observed angiogenesis, cell migration, tumor cell intravasation, and responses to drugs administered through the vascular network. Siler et al. demonstrated a gut-on-a-chip model with intestinal vasculature using patient-derived intestinal ECs, epithelial cells, and subepithelial myofibroblasts [173]. Using the model, intestinal subepithelial myofibroblasts were found to play an essential role in angiogenesis, and also an anti-cancer agent, Erlotinib, showed anti-angiogenic properties. A patient-specific, microfluidic neuroblastoma model was recently developed by Nothdurfter et al. using bioprinting techniques [174]. Specifically, ECs were seeded in bioprinted channels, in which microvasculature was formed within a GelMA/fibrin-based matrix that contained patient-derived neuroblastoma spheroids. As a result, neuroblastoma spheroids promoted microvascular formation into the spheroids, and an anti-cancer drug (i.e., bortezomib) was highly efficient at inhibiting angiogenesis.

In summary, microphysiological systems engineered with patient-specific cells have already proven to be a valuable tool to facilitate personalized medicine in the future. Continuing efforts and further developments in this field will be key to further enhancing the understanding of inter-individual differences in microvascular formation (i.e., vasculogenesis) at biological and molecular levels. Doing so will ultimately enable clinicians to optimize and cater treatments to individual patients.

## 3. Vascular Genesis and Angiogenesis for Tissue Engineering and Regenerative Medicine

Therapeutic vascular genesis and angiogenesis present an interesting approach in regenerative medicine, for example, in the treatment of ischemic diseases. Ischemia, or a restriction of oxygen supply to a tissue, has various etiologies, such as cardiovascular and hematologic diseases, trauma, graft failure/rejection after organ transplantation, etc. Between these etiologies, cardiovascular ischemia is the most common and presents in many clinical forms, of which the most critical are summarized in Table 2.

The common denominator in the revascularization options is ‘time is tissue’. The sooner an arterial occlusion is resolved, the more chance the ischemic tissue has to recuperate and survive. Delayed revascularization often leads to irreversible damage and tissue loss.

Nonetheless, reperfusion of the ischemic tissues is not the end of the story. Essential in the treatment of ischemia is to also address potential ischemia–reperfusion (I/R) injury. This pathophysiological phenomenon exists of an exacerbated inflammatory response after reperfusion/revascularization, which injures the ischemic tissue even more. Hitherto, therapeutic initiatives to counter I/R injury have not been successful in a clinical setting [182]. Thus, patients at risk for critical ischemic diseases could benefit from angio/vasculogenesis-stimulating treatments in a secondary (i.e., prevention in chronic ischemia) or tertiary (after reperfusion) setting.

One novel treatment option is to manipulate the angiogenesis program, such as the VEGF, angiopoietin, and FGF pathways. These pathways are crucial for EC proliferation, migration, survival, and the stabilization of formed vascular networks [183]. Unfortunately, the administration of angiogenic cytokines as recombinant protein or gene therapy has not yet succeeded in clinical trials, as reviewed in [184,185,186].

Furthermore, vasculo- and angiogenesis research potentially has far more applications than ischemic disease treatment. As mentioned earlier, an integrated mature vascular network is indispensable for any relevant healthy tissue and cancer model, both for in vitro studies and translation into in vivo models. This means the vascularity needs to develop in an integrative manner with other cell types, avoiding the creation of a static coculture [187]. In this regard, each tissue requires its own specific vascular network. Anatomical organ-specific vascularity has been established for a while, but only recently have studies focused on the structural and molecular differences. These studies demonstrate the heterogeneity in endothelial structure in bone, brain, cardiac, liver, kidney, gut, skin [188], and pancreatic [30] capillaries. Where large vessels, brain, lung and heart tissue present a continuous endothelium with strong adherence junctions [189], the lung, kidney and gut are built of fenestrated endothelium. Even more permeability is allowed in discontinued endothelium, such as in liver tissue. Compared to structural differences, molecular heterogeneity is understudied in humans due to the lack of access to multi-organ samples. Existing studies are based on murine samples, showing a large variety of angiogenic and endothelial markers between different organs [31,190]. It seems important to take these organ-specific structural and molecular differences into account when assessing vascular development in engineered constructs since the vascular structure often plays a key role in organ function.

To create these organ-specific vasculatures, researchers mostly look at autologous PSC and EPC as cell sources due to their proliferation and endothelial differentiation potential, their accessibility and neutral immunogenic status. PSC can be differentiated into ECs in three different ways [191]. Firstly, through self-aggregation in suspension and the formation of embryoid bodies (EBs). Secondly, endothelial differentiation can be stimulated by coculturing PSC with stromal cells (so-called feeder cells) [192]. Thirdly, and currently most applied, is seeding PSC on substrate-coated (e.g., Matrigel, gelatin, fibronectin) culture plates and subsequentially adding recombinant growth factors [193,194]. After differentiation, ECs tend to form tube-like structures quite easily. However, functional vessels that support graft viability need to be perfusable, as discussed in the next section.

### 3.1. Strategies for Achieving Vessel Perfusability within In vitro Capillary Networks

Tissue viability is directly correlated with the perfusability of the entire vascular tree. Two regions of interest are the anastomoses on a capillary (10–50 µm) and venous/artery (>0.3 mm) level. On the capillary level, oxygen and nutrients are exchanged for carbon dioxide and waste products, continuously in equilibrium due to the perfusion in these microvessels. Pioneered by the Jeon group, in vitro perfusable microvessels based on endothelial cell self-assembly in fibrin gel-based ECM have been widely used in the field of vascular engineering [86]. George and Hughes’ groups further developed this technique to allow co-culture with tumor cells to create 3D microtumor models supported by lumenized perfusable vessels [195]. This work introduced a novel method of laminin coating to support vascular perfusion. Chiu et al. (2012) separated a murine artery and vein with a micropatterned PDMS and Tβ4-loaded collagen hydrogel (Figure 5A,B). EC outgrowth was followed up for 21 days, and perfusability was demonstrated by fluorescent dextran injection [196]. Kamm’s group went a step further and showed that in vitro microvessels are perfusable and can be applied for studying tumor and immune cell extravasation [197,198,199]. Recently, this group demonstrated that higher vessel perfusability could be achieved by mean of seeding one highly concentrated endothelial-cell-contained fibrin layer outside of the gel channel with normal endothelial density (Figure 5C), resulting in a larger opening of microvessels to the media channels and, thus, higher perfusability (Figure 5D) [200]. Mori et al. (2020) engineered a perfusable vascular network in a liver construct (Figure 5E–H). They started from a tubular design and embedded it in HepG2-MSC-HUVEC populated collagen. After perfusion, capillary growth was observed inside the hydrogel with a connection to the main channel. Fluorescent images show a sinusoid-like morphology and integration of the ECs in the liver construct. Functionality was demonstrated by Indian Ink perfusion and albumin measurements on the conditioned medium [201]. Furthermore, Bonanini et al. (2022) show that perfusable capillaries can be applied in high-throughput assays (Figure 5I–L). Their OrganoPlate holds 64 microvessel units that fit underneath a 384-well plate. HUVEC-monoculture and collagen I-hydrogel were sufficient to create a perfusable network. Additionally, their setup allows vascular interaction with organoids and spheroids and can be applied to study tissue regeneration or as a high-output drug testing platform [202]. In another study, a versatile platform based on perfusable vascular capillaries is demonstrated by Paek et al. (Figure 5M–P). Their lung fibroblast–HUVEC-fibrin vascular bed is applied for vascular adipose tissue engineering as a model for the blood–retinal barrier, a vascularized lung tumor, and vascular inflammation [203]. All scenarios showed a more in vivo-like phenotype compared to static co-cultures. In other studies, self-assembled microvasculatures for the reconstruction of the BBB can also be performed by co-culturing brain ECs, astrocytes, and pericytes, or human mesenchymal stem cells in a fibrin-based extracellular matrix [204,205]. In short, all these different studies display functional vasculo- and angiogenesis, which is essential for tissue viability and function.

### 3.2. Vascular Perfusion within Engineered 3D Tissues

The diffusional limit for oxygen and nutrients is the main factor limiting the survival and growth of engineered 3D tissues [206]. By incorporating channels that are generally less than 100–200 microns away from each other, the volume that is out of reach for diffusion from the closest vascular branch, i.e., beyond the radius of Krogh’s cylinder, is minimized [61,207,208]. Table 1 briefly overviews the additive manufacturing tissue engineering methods to biofabricate vasculature with additional tissue compartments. Moreover, by adding a culture medium fluid flow through the biofabricated macrovasculature, the diffusion of oxygen and nutrients is aided by convective mass transport: If there is sufficient flow, an axial gradient of oxygen and nutrients is prevented [44,47,50,206]. Kinstlinger et al. show that a perfused vascular network branching into daughter vessels on a distance smaller than Krogh’s cylinder radius is able to provide a viable amount of oxygen to a large construct (6.5 mL hydrogel volume), each containing 65 million hepatocytes and 32.5 million fibroblasts [44].

In addition to improving the survivability of biofabricated organs, the introduction of vasculature and flow culture has an important role in fulfilling the native function of the intended organ. 

In 2019, the group led by Jennifer Lewis reported a 3D vascularized proximal tubulus model [57]. This model is composed of two adjacently bioprinted channel structures, one of which is lined with ECs and the other with epithelium, see Figure 6A. Two separate flow systems were used to perfuse the vasculature and proximal tubulus. For albumin and glucose, they proved that the active reabsorption of the vascularized kidney model is similar to the native proximal tubules [57].

Recently, the group led by Dong-woo Cho used their coaxial bioprinting method to directly bioprint a 3D microfluidic vascularized renal tubular tissue-on-a-chip. This construct also showed in vivo-like behavior, such as barrier function and albumin reabsorption. Additionally, they introduced a complex tubular construct with both mono- and bilayer structures to recreate glomerular microvessels with afferent and efferent monolayer arteriole structures and glomerular capillaries designed to contain an inner layer with glomerular ECs and an outer layer lined with podocytes, see Figure 6B [74].

Byambaa et al. utilized an indirect bioprinting method to introduce microvasculature in a bone tissue construct. In this construct, a central vascular channel was surrounded by silicate nanoplatelets loaded with GelMA hydrogel to induce osteogenesis in the encapsulated mesenchymal stem cells. The osteogenic activity was significantly upregulated for the constructs cultured with a culture medium flow compared to a static culture [65].

In 2019, the Miller group reported a 3D vascularized lung model created by stereolithographic patterning. As depicted in Figure 6C, a distal lung unit was designed by the fusion of spheres connected to a shared air duct. To design the vascular architecture, the alveolar design was enlarged around the alveoli and transformed into a covering mesh containing 354 vessel segments and 233 fluidic branch points. The hydrogel structure (20 *w*/*v*%, 6-kDa PEGDA) was able to withstand more than 10,000 ventilation cycles (24 kpa, 0.5 hz) during 6 h of RBC perfusion. The ventilation cycles encompassed a switch between humidified oxygen and nitrogen. A drop in the oxygenation of RBCs was visible during the nitrogen ventilation and was restored during oxygen ventilation, which signifies that their model fulfills the main function of a lung. This group also showed that the interstitial space could be populated with lung fibroblasts and the airway with epithelial cells [54].

Evidently, 3D biofabricated constructs with in vivo-like morphology and function are promising; however, in vivo validation is crucial. One biofabrication technique that has shown to be successful in vivo is to change the pore size in the construct. Increasing the pore size has been shown to facilitate vascular ingrowth, and Bai et al. suggested a minimum of 400 µm [209].

Additionally, promising is the addition of proangiogenic factors to the construct. Liu et al. applied laser sintering to create a porous 3D polycaprolactone/hydroxyapatite (PCL/HA) scaffold for bone regeneration. Next, they loaded the scaffold with VEGF and implanted this in a cranial defect rat model. Over time, the VEGF-loaded scaffold demonstrated enhanced blood vessel formation and bone regeneration over the VEGF-free control [210].

Another promising technique that has been discussed in this review is the prevascularization of constructs. Kuss et al. created a prevascularized construct with mesenchymal stem cells and HUVECS and generated capillary-like networks within a porous 3D-printed PCL/hydroxyapatite scaffold. This prevascularized construct showed higher vessel density 4 weeks after subcutaneous implantation compared to constructs without prevascularization [211]. Similarly, Redd et al. prevascularized collagen scaffolds with hESC-derived ECs and cardiomyocytes. The scaffolds were implanted in infarcted hearts 4 days after ischemia/reperfusion. They observed a higher cardiomyocyte and perfused blood vessel density in the constructs with a hierarchical prevascularization, with both macro- and microvessels [212]. Additionally, for cardiac repair, Jang et al. created a prevascularized 3D-printed patch with cardiac progenitor cells, mesenchymal stem cells and VEGF in a cardiac ECM ink [213]. Implanted in an ischemic heart rat model, the patch induced neovascularization and even improved cardiac functions after 8 weeks. Recently, the Jeon group has grown vascularized tissues between two parallel polyethylene glycol diacrylate membrane for easy detachment and subsequential implantation into an ischemic hind limb mouse model to recover blood perfusion [214]. Other prevascularization initiatives in 3D printing, their challenges and opportunities in in vivo translation are elegantly summarized by Joshi et al. in [215].

### 3.3. Anastomosis of Graft Biofabricated Tissues

For perfusion on a macroscopic level, engineered tissues require a couplet of vessels (arterial and venous) that allow surgical anastomosis, connecting the engineered vascular network with the in vivo cardiovascular system. Today, super-microsurgical techniques allow microscope-assisted anastomosis of vessels of 0.3–0.8 mm [216,217]. However, these anastomoses are surgically challenging. Moreover, the smaller the vessels, the more prone they are to thrombogenic occlusion. For example, synthetic Dacron (polyethylene terephthalate) vascular grafts >8 mm have shown to be successful in the clinic [218,219], while smaller vessels (<6 mm) occluded due to surface thrombogenicity and intimal hyperplasia [218]. Surface thrombogenicity is caused by the absence of an endothelial lining of the synthetic vessel. To reduce this thrombogenic risk, anticoagulants such as heparin [220], hirudin, dipyridamole [221], and non-thrombogenic phospholipid polymers [222] can be embedded in these grafts. A second option is to seed the lumen with anti-thrombotic cells such as ECs, EPC, or mesenchymal stem cells [223]. The latter approach is more integrative and may offer more opportunities to link the macrovessels with the capillary networks, recreating the entire vascular tree. Levenberg and colleagues applied a similar approach. They applied bioprinting techniques to create an anastomosable vessel surrounded by self-assembled capillaries. The included cell types were human adipose tissue microvascular ECs (HAMECs) and dental pulp stem cells. Next, the construct was anastomosed with a rat femoral artery [224].

Other groups play with bioprinting concepts for clinical applications, as summarized in 2020 by Wu et al. [225]. Briefly, most studies use intraoperative bioprinting for skin and bone engineering. Just two of the seven described studies included ECs, and only these studies assess vascularization in the bioprinted constructs.

## 4. Vascularized Engineered Tissue: Challenge and Perspective

Despite numerous challenges in vascularized tissue engineering, developments in the field are progressing rapidly. In consideration of functional tissue replacement by mimicking the anatomically relevant blood vessel with biological functionality, understanding its various requirements is essential. The major challenges in vascular graft design include recapitulation of diverse sizes, cellular compositions, and structural variations of organ-specific vasculatures. Moreover, the achievement of the vascular density within the engineered tissues similar to in vivo requires the co-development of vasculatures and functional cells.

The manufacturing techniques used to fabricate vascularized tissue must be able to produce blood vessels with maximum cell survival and ideal recapitulation of the three essential vascular layers, including outer layer adventitia for structural support, middle layer for elastic support, and inner layer with endothelial lining, which are the essential parts to recapitulate vascular tissue. Currently, one of the biggest challenges in reproducing blood vessels is how to distribute the cells within the vascular tissue to recapitulate the natural arrangements [12]. Inside the blood vessels, the middle layer is composed of tunica media, which includes elastic and muscular tissue such as SMCs and ECs, while the inner vascular layer has tunica intima to regulate the friction to the blood flow in contact. Several vascular tissue manufacturing techniques, including the acellular scaffold-based fabrication process, have the static method of seeding cells in common [226]. This method of seeding cells can only achieve a suboptimal distribution since the SMC cannot be placed directly in the middle of the vascular layer. This method also fails while achieving homogeneous distribution of ECs on the luminal vascular layer.

Engineered vascularized tissue is reproduced using different strategies, of which neoangiogenesis is widely used. This strategy mainly focuses on tubular structures and their properties, which can be achieved by different fabrication approaches such as biologically derived vessel systems and systematically manufactured tubular systems. These strategies have angiogenic growth factors as a common element to activate the endothelial (progenitor) cells, promote cell assembly, and help vessel formation and maturation. Despite the fine integration of biological molecules, in particular collagen, fibronectin, and cell adhesion peptides (RGD), fibronectin heparin binding over the scaffold surface, these fabrication methods are solemnly not efficient in achieving the desired three-layered vascular tissue structure.

The blood vessel must be able to withstand the stitching during in vivo implantation and endure blood pressure inside the body. From the above-mentioned perspective, it is clear that reproducing cellular constructs recapitulating the composition of the blood vessel might not be enough and that these constructs shall be conditioned/matured within the dynamic bioreactors. In future studies, these active maturation conditions may be the final key to obtaining functional implantable blood vessels.

The building of a biocompatible surface that encounters blood has always been a major challenge in fabricating vascular tissues [12]. While reproducing these surfaces, it is necessary to imitate anti-thrombogenic mechanisms for the successful attainment of a functional blood vessel. Although the structural complexity and the composition of the blood vessel’s different layers are hard to achieve, the use of specifically designed biomaterials, cells, and biological molecules could help us meet this challenge. However, these technological signs of progress are still unable to produce perfect vascular tissue for all patients. Currently, most of the engineered vascular tissue studies are composed of biomaterials with only a single layer of ECs. On the other hand, real blood vessels are composed of ECs as well as SMCs which ensures permeability and mechanical strength. Presently, blood vessel manufacturing techniques, such as cell sheet rolling, cell modeling, and electrospinning have reached their limits. Thus, the production of vascular tissue remains a major challenge for tissue engineering. The defects, such as a lack of homogeneity in cell distribution, the poor mechanical strength of the constructs, limited lifespan, and the limitations of the exact representation of the three cellular layers in the cylindrical form, open a pathway for the emerging 3D bioprinting manufacturing technology. Finally, the most important parameter to consider is the host’s immune response to the implanted blood vessel. To overcome this issue, in-depth research must be conducted to better understand stem cell behavior in 3D culture and biomaterial–blood interaction.

## Figures and Tables

**Figure 1 micromachines-14-00978-f001:**
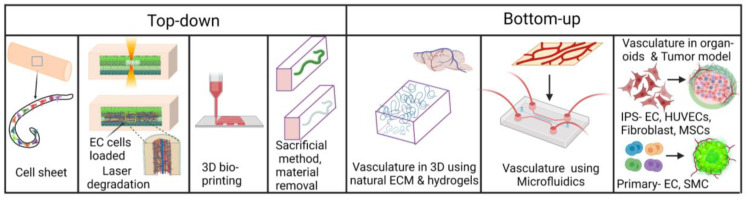
Methods for creating in vitro vasculatures.

**Figure 2 micromachines-14-00978-f002:**
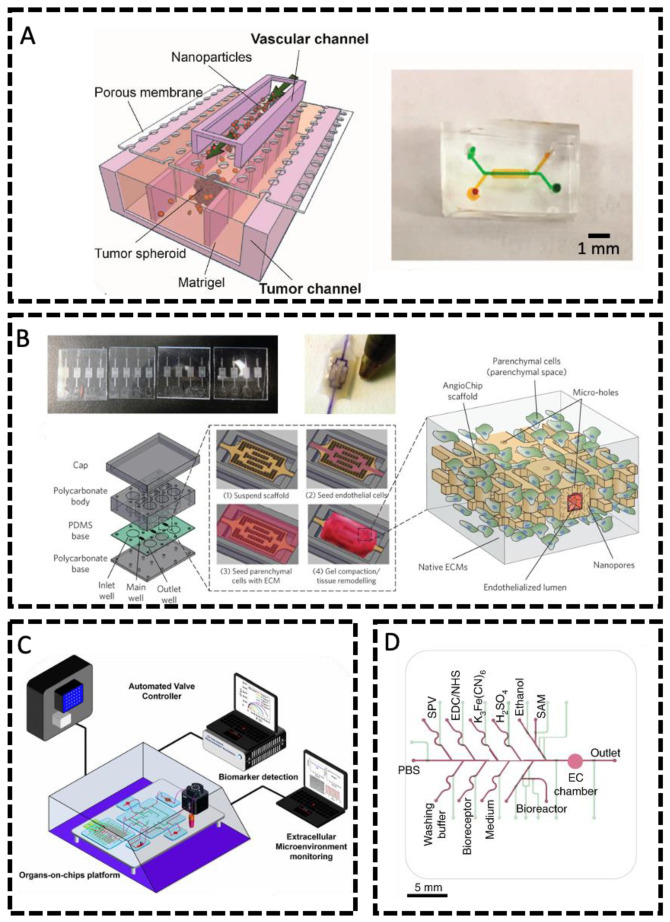
Fluidic flows, pumps, and sensor integration. (**A**) Schematic illustration showing the PDMS double-layer system, where a porous membrane is sandwiched between vascular layers channel and bottom tumor layers. The image on the right shows the actual device. Reprinted with permission from Wang et al. [92]. Copyright 2018 American Chemical Society. (**B**) Digital image showing the model and the actual device (right image) using the biodegradable elastomer poly(octamethylene maleate (anhydride) citrate. The bottom panels show the schematic representation of the layering and construction of the barrier of the chip. Reproduced with permission from Springer Nature [96], copyright 2016. (**C**) Image showing the schematic layout of the automated monitoring and control of organoids with electrochemical and physical sensors integrated OOC cascade. Fluids are controlled using a peristaltic pump inside an incubator. (**D**) Schematic image showing the liquid flow of the regeneratable affinity-based biosensor integrated organ-on-a-chip (OOC) device monitoring glutathione S-transferase-alpha (GST-α) and creatine kinase-MB (CK-MB). Image recreated with permission from Zhang et al. [99]. Copyright 2017 National Academy of Sciences.

**Figure 3 micromachines-14-00978-f003:**
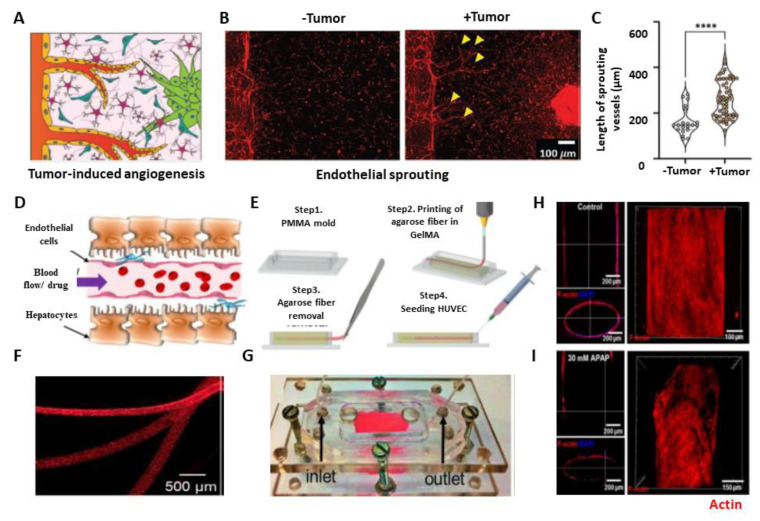
Vascular angiogenesis on a chip. (**A**) Schematic of tumor-induced angiogenesis. (**B**) New vessel formation (yellow) from pre-existing vessel toward the GBM spheroid. Scale bar: 100 μm. (**C**) Length and number of neo-vessels toward the GBM spheroid. Significance is indicated by **** for *p* < 0.0001; all by unpaired *t*-test). Image A to C are recreated with permission from Seo et al. [113]. Copyright 2022, Wiley-VCH GmbH. (**D**) Schematic illustration of a vascularized liver construct. (**E**) 3D bioprinting to engineer 3D liver tissue construct with central vessel using agarose. (**F**) Microchannels embedded liver construct. The red dye solution was perfused into the hydrogel construct after molding and crosslinking. (**G**) Photographs showing the microfluidics that enable perfusion in the bioreactor. Confocal microscopy images of vasculature construct showing: (**H**) without APAP (control) where the HUVEC layer remains intact and (**I**) with 30 mM of APAP exhibit a damaged vessel. Images D to I are reproduced from Massa et al. [122] with the permission of AIP Publishing.

**Figure 4 micromachines-14-00978-f004:**
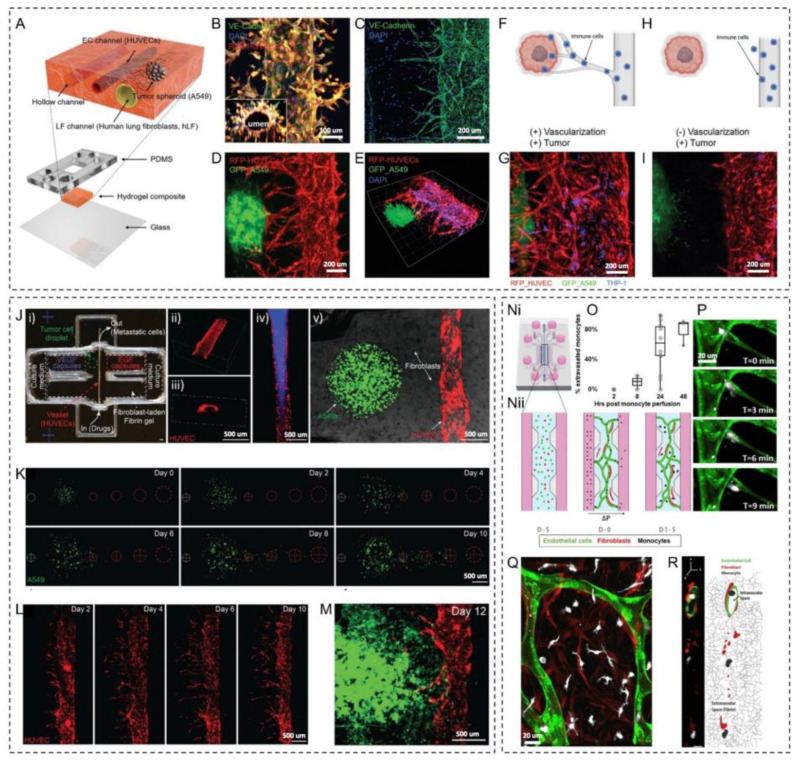
Three-dimensional vascularized tumor in vitro systems to model intravasation, extravasation, and invasion. (**A**) Illustration of the 3D lung cancer model that recapitulates blood vessels within a hydrogel composite. (**B**) Microscopic images that highlight the constructed blood vessels (red: HUVECs that express RFP; green: VE-cadherin; blue: DAPI). (**C**) A 3D image of the sprouts grown from the main HUVEC blood vessels. (**D**) A projection of the 3D vascularized lung cancer model (red: RFP-HUVEC and green: GFP-A549). (**E**) A 3D reconstructed view (red: RFP-HUVEC, green: GFP-A549, and blue: DAPI). (**F**) An illustration of a vascularized tumor, in which the movement of immune cells through the small blood vessels is enhanced. (**H**) An illustration of a tumor without vascularization, in which the movement of immune cells is restricted. (**G**,**I**) More THP1 cells were retained in the area closed to the tumor spheroid in the vascularized than nonvascularized tumor model when THP1 cells are perfused into the main blood vessel (red: RFP-HUVEC, green: GFP-A549, and blue: THP1). Reproduced with permission [8]. Copyright 2022, Wiley-VCH GmbH. (**J**) Metastatic model. (**i**) A photograph of a 3D-printed container used for testing the spread of tumor cells. (**ii**,**iii**) Three-dimensional images of the top view (upper panel) and cross-section (lower panel) of a small blood vessel lined by HUVECs within a gel made of fibrous protein, showing the interior of the vessel. (**iv**) Microscopic images showing a vessel filled with fluorescent liquid (blue, originally poly(fluorescein isothiocyanate allylamine hydrochloride)). (**v**) A composite image showing a representative model of a tumor before laser-triggered rupture of capsules containing EGF and VEGF (green fluorescence: A549s that express GFP, red fluorescence: HUVECs that express RFP, bright field: fibroblasts). (**K**) A series of microscopic images showing the distribution of A549 cells that express GFP over time, demonstrating guided migration (red circles: EGF capsules; white circle: control capsule without growth factor loading; and cross lines: laser rupture pathways). (**L**) A series of microscopic images showing small blood vessels that sprout from the main vessel and extend in a single direction over time, indicating guided sprouting angiogenesis by VEGF capsules. (**M**) A microscopic image of a metastatic model on day 12, showing that A549 cells move towards and enter the blood vessels through the gel made of fibrous protein and fibroblast cells (green channel: A549 cells that express GFP; red channel: HUVECs that express RFP). Reproduced with permission [138]. Copyright 2019, Wiley. (**N-i**) A diagram of the microfluidic device, showing the central compartment shaded in blue that has the cells surrounded by the small channels filled with liquid (pink). (**N-ii**) A close-up of the central region that contains the cells mixed in a 3D hydrogel at different time points. (**O**) Measurement of the movement of monocytes out of the blood vessels after the introduction of monocytes in the network of small blood vessels. (**P**) 3D images of a monocyte (white) moving through the cells that line the blood vessels (green). (**Q**) A representative image of the monocytes 4 days after the introduction. (**R**) A representative cross-sectional view of a segment of a small blood vessel (green) and the area outside of it, as observed with 3D microscopy and represented in a diagram (right panel). Fibroblast cells (red) are found in the area outside of the 3D matrix made of fibrous protein, while monocytes (white) can be found either inside the hollow vessels or outside of the 3D matrix made of fibrous protein. The scale bar is 10 μm. Reprinted from Biomaterials 198 (2019): 180-93. “The Effects of Monocytes on Tumor Cell Extravasation in a 3d Vascularized Microfluidic Model.” Boussommier-Calleja, A., Y. Atiyas, K. Haase, M. Headley, C. Lewis, and R. D. Kamm. [140]. Copyright 2019, with permission from Elsevier.

**Figure 5 micromachines-14-00978-f005:**
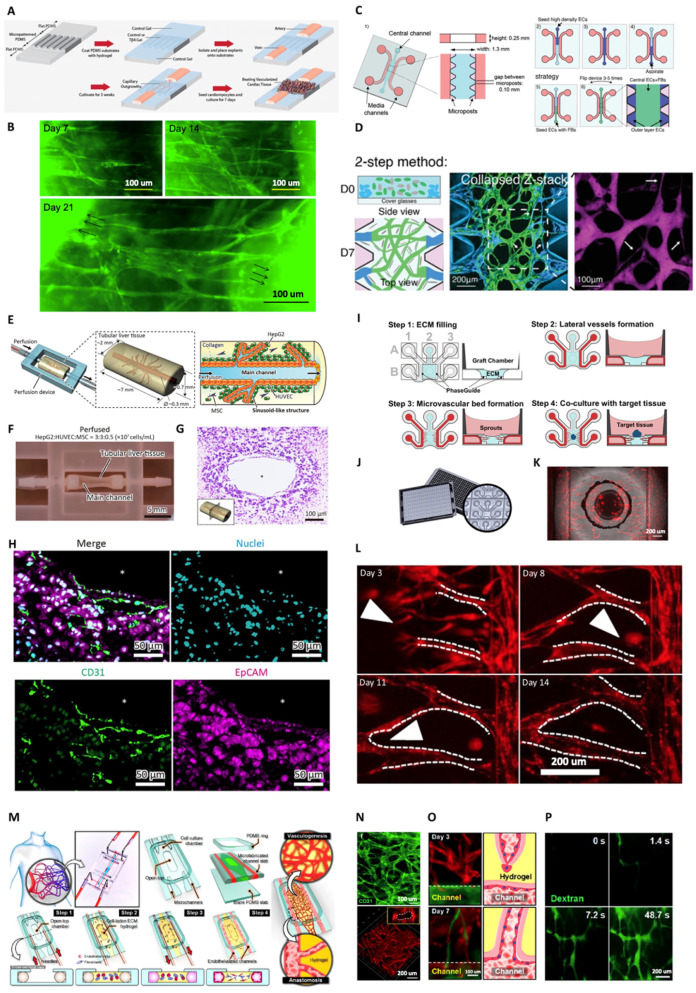
Engineered perfusable capillary network. (**A**,**B**) Design of PDMS chip and perfusable vascular network, arrows depict anastomoses. Adapted with permission from Chiu et al. [196]. Copyright 2012 National Academy of Sciences. (**C**,**D**) New strategy to create in vitro perfusable microvasculature using 2 different gel densities. (**C**) From 1 to 6: two-steps seeding strategy. (**D**) Two differently labeled HUVECs (blue and green) with different seeding densities form perfusable microvasculature networks together. Dextran70 kDa in magenta. Adapted with permission from Wan et al. [200]. Copyright 2022 Wiley. (**E**–**H**) Perfused tubular liver construct with (**G**) a mature central lumen. (**H**) Immunofluorescent images in which cells were stained for nuclei (cyan), CD31 (for HUVEC, green), and EpCAM (for HepG2, magenta), showing the integration of capillaries in liver tissue. Reproduced with permission from Springer Nature [201]. (**I**–**L**) High-throughput vascular assay for spheroid and organoid interaction studies. (**L**) Fluorescent images of maturing capillaries (RFP-labeled HUVEC). Reproduced with permission from Springer Nature [202]. (**M**–**P**) Design of multifunctional microvascular chip. (**N**) CD31-staining showing dense endothelial network. (O) Anastomoses between side channels and central vascular plexus. (**P**) Perfusion of the network with fluorescent-labeled dextran, adapted with permission from Huh et al. [203]. Copyright 2019 American Chemical Society.

**Figure 6 micromachines-14-00978-f006:**
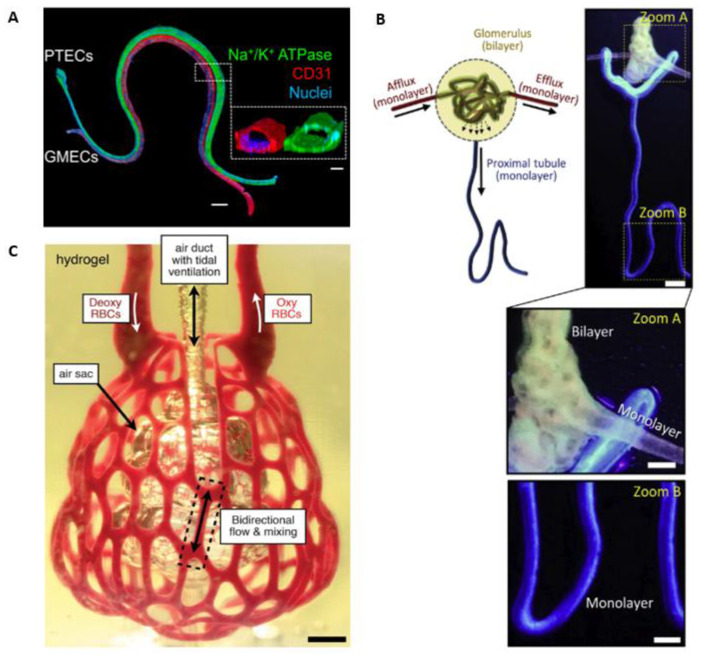
Vascular perfusion in engineered tissues. (**A**) Overview and frontal view of the vascularized proximal tubulus model, according to Lin et al. Green = Na+/K+ ATPase, red = CD31, blue nuclei (NucBlue staining) and scale bar = 1 mm. Inset: Cross-sectional image of the separate channels lined with PTECs and GMECs. Scale bar = 100 µm. Adapted with permission from [57]. Copyright 2019 National Academy of Sciences. (**B**) Illustration depicting the structure of glomerulus and proximal tubulus in the native kidney next to a 3D bioprinted renal structure consisting of monolayer and bilayer tubular structures. Scale bar overview image = 500 µm, scale bar zoom A and B = 250 µm. Reprinted from Biomaterials 232 (2020): 119734. “Three-Dimensional Cell-Printing of Advanced Renal Tubular Tissue Analogue.” Singh, N. K., W. Han, S. A. Nam, J. W. Kim, J. Y. Kim, Y. K. Kim, and D. W. Cho [74], with permission from Elsevier. (**C**) Image of the biofabricated distal lung unit, while the alveoli are ventilated with O_2_ and RBC, are perfused through the surrounding vasculature. Color-filtered views (not shown) of the RBC perfusion show that ventilation creates a bidirectional flow with mixing of RBC, which is considered beneficial for oxygenation. Scale bar = 1 mm. Adapted with permission from Grigoryan et al. [54]. Copyright 2019 Science.

**Table 1 micromachines-14-00978-t001:** Engineered tissues with vasculatures and their fabrication method.

Manufacturing Method	EC	Associated Cell Type(s)	Desired Organ	Biomaterial	Fugitive Ink	Flow and/or Static Culture	Microvascular Integration	Ref.
Conventional manufactured sacrificial templates (CMST): Annular mold	Bovine aortic ECs	Bovine smooth muscle cells, and adventitial fibroblasts	Only vasculature	Collagen	/	Static	No	Weinberg & Bell (1986) [45]
CMST: Sheet rolling	Human umbilical vein endothelial cells (HUVECs)	Human vascular smooth muscle cells and human skin fibroblasts	Only vasculature	Cell sheets produced by smooth muscle cells and fibroblasts	/	Flow	No	L’heureux et al. (1998) [11]
Additive manufactured sacrificial templates (AMST): Stereolithography (SLA)	/	Red blood cells, Human lung epithelial cells Additionally, Human lung fibroblasts	Lung	6 kDa Polyethylene glycol diacrylate (PEGDA) or a mixture of methacrylated gelatin (GelMA) and 6 kDa PEGDA	6 kDa PEGDA or mixture of GelMA and 6 kDa PEGDA	Flow	No	Grigoryan et al. (2019) [54]
AMST: Stereolithography (SLA)	HUVECs	Human colorectal adenocarcinoma cells	Vasculature and associated single cells	PEGDA: 0.7 kDa Additionally, GelMA added after templating	PEGDA: 0.7 kDa	Flow and static	No	Zhang et al. (2017) [48]
AMST: Selective Laser Sintering	/	Hepatic aggregates consisting of primary rat hepatocytes and human dermal fibroblasts	Liver	Agarose	Isomalt and cornstarch	Flow and static	No	Kinstlinger et al. (2020) [44]
AMST: Selective Laser Sintering	HUVECs	IMR—90 lung fibroblasts	Vasculature and associated single cells	GelMA	Isomalt and cornstarch	Flow and static	No	Kinstlinger et al. (2020) [44]
AMST: 3D printing	HUVECs	C3H/10T 1/2 cells and Human Embryonic Kidney cells	Vasculature and associated single cells	Fibrin	Carbohydrate glass	Flow and static	Yes	Miller et al. (2012) [47]
AMST: 3D (Bio)printing	Human adipose microvascular ECs	Dental pulp stem cells	Tissue flap, transplantable with direct anastomosis	PLLA–PLGA blend and recombinant human collagen methacrylate	butanediol vinyl alcohol copolymer	Static	Yes	Szklanny et al. (2021) [55]
AMST: 3D (Bio)printing	HUVECs	Mouse calvarial pre-osteoblasts cells	Vasculature and associated single cells	GelMA	Agarose	Static	No	Bertassonia et al. [56]
AMST: 3D (Bio)printing	Glomerular microvascular ECs	Proximal tubular epithelial cells	Kidney	Gelatin-fibrin blend	Pluronic F-127 with high-molecular weight PEO	Flow	No	Lin et al. (2019) [57]
Indirect bioprinting	HUVECs	human neonatal dermal fibroblasts (HNDFs)	Vasculature and associated single cells	GelMA	Pluronic F-127	Static	No	Kolesky et al. (2014) [51]
Indirect bioprinting	/	Rabbit bone marrow-derived mesenchymal stem cells	Only vasculature	Sodium alginate, medium viscosity	polycaprolactone	Static and flow	No	Lee et al. (2018) [58]
Indirect bioprinting	/	Sheep primary bone marrow stromal cells	Vasculature and associated single cells	Silanized Hydroxypropylmethylcellulose	Gelatin	Static and flow	No	Figueiredo et al. (2020) [59]
Indirect bioprinting	HUVECs	/	Only vasculature	Collagen	Gelatin	Static and flow	Yes	Lee et al. (2014) [50]
Indirect bioprinting	HUVECs	Human lung fibroblasts	Vasculature and associated single cells	Collagen and fibrin	Gelatin	Static and flow	Yes	Lee et al. (2014) [60]
Indirect bioprinting	HUVECs	Human bone-marrow-derived mesenchymal stem cells (hMSCs) and human neonatal dermal fibroblasts (HNDFs),	Bone	Fibrin-gelatin blend	Pluronic F-127	Flow	No	Kolesky et al. (2016) [61]
Indirect bioprinting	HUVECs	Normal Human Dermal Fibroblasts and Human umbilical artery smooth muscle cells (HUASMC)	Only vasculature	Fibrin and Fibrin-collagen blend	Gelatin	Flow	No	Schöneberg et al. (2018) [62]
Indirect bioprinting	HUVECs	Human mesenchymal stem cells	Vasculature and associated single cells	Methacrylated alginate and methacrylated hyaluronic acid (HAMA)	Pluronic F-127	Static	No	Ji et al. (2019) [63]
Indirect bioprinting	HUVECs	HepG2, Human foreskin fibroblasts, and human umbilical cord MSCs	Liver, transplantable with direct anastomosis	GelMA–fibrin blend	Gelatin	Flow	Yes	Liu et al. (2021) [64]
Indirect bioprinting	HUVECs	Bone-marrow-derived human mesenchymal stem cells	Bone	High-stiffness GelMA	Low stiffness GelMA	Flow	Yes	Byambaa et al. (2017) [65]
Indirect bioprinting	HUVECs	Human dermal neonatal fibroblasts	Only vasculature	Gelatin–poly(ethylene glycol)–tyramine mixed with horse radish peroxidase	Gelatin −H_2_O_2_ blend	Static	No	Hong et al. (2019) [66]
Indirect bioprinting	HUVECs	/	Only vasculature	Pluronic F-127-BUM (with collagen 1 additive)	Pluronic F-127	Static	No	Millik et al. (2019) [67]
Indirect bioprinting	HUVECs	/	Only vasculature	Sodium alginate	CaCl_2_	Static	No	Attalla et al. (2016) [68]
Indirect bioprinting	HUVECs	Human mesenchymal stem cells	Only vasculature	GelMA-Alginate—4-arm poly(ethylene glycol)-tetra-acrylate (PEGTA) blend	CaCl_2_	Static	No	Jia et al. (2016) [52]
Indirect bioprinting	/	Primary human umbilical vein smooth muscle cells	Only vasculature	Sodium alginate	CaCl_2_	Flow and static	No	Zhang et al. (2015) [69]
Direct bioprinting	HUVECs	Human Coronary Artery Smooth Muscle Cells and Human bone marrow-derived mesenchymal stem cells	Vasculature and associated single cells	Catechol-functionalized GelMA	Pluronic F-127	Flow and static	No	Cui et al. (2020) [70]
Direct bioprinting	HUVECs	/	Only vasculature	Vascular-tissue-derived dECM–sodium alginate blend	Pluronic F-127—CaCl_2_ Blend	Static	No	Gao et al. (2017) [71]
Direct bioprinting	HUVECs	Human HL-60 cell line	Vasculature and airway inflammation modelling	Vascular-tissue-derived decellularized ECM–sodium alginate blend	Pluronic F-127—CaCl_2_ Blend	Flow and static	Yes	Gao et al. (2018) [72]
Direct bioprinting	HUVECs	Human aortic smooth muscle cells	Only vasculature	Porcine aorta derived decellularized ECM–sodium alginate blend	Pluronic F-127—CaCl_2_ Blend	Flow and static	No	Gao et al. (2019) [73]
Direct bioprinting	HUVECs	Renal proximal tubular epithelial cells	Kidney	Kidney-derived decellularized ECM–sodium alginate blend	Pluronic F-127—CaCl_2_ Blend	Flow and static	No	Singh et al. (2020) [74]
Multimaterial SLA	HUVECs	Human bone marrow-derived mesenchymal stem cells and human dermal fibroblasts	Only vasculature	GelMA	HAMA	Static	Yes	Orellano et al. (2022) [75]

**Table 2 micromachines-14-00978-t002:** Critical cardiovascular ischemic diseases and current revascularization options per organ.

Organ	Disease	Treatment	Ref
Heart	Acute coronary syndrome	- Percutaneous coronary intervention (balloon/stent) or coronary artery bypass graft - Antithrombotic therapy	[175]
Brain	Ischemic stroke	- Intravenous thrombolysis - Mechanical thrombectomy	[176]
Limb	Critical limb ischemia	- Endovascular angioplasty and stenting - Open angioplasty, endarterectomy or bypass	[177,178,179]
Bowel	Acute mesenteric occlusion	- Open embolectomy or bypass - Percutaneous thrombus aspiration, angioplasty and/or stenting	[180,181]

## Data Availability

No new data was created or analyzed in this study and data sharing is not applicable to this article.

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
