# Peer review of "Engineered Vasculature for Cancer Research and Regenerative Medicine"

_micromachines, 2023, doi:10.3390/mi14050978_

Round 1
Reviewer 1 Report
This manuscript is well-written, and fruitful in the field of organ on-a-chip and regenerative medicine.
I suggest accepting this paper after minor revision.
Minor comments) Scale bars in Figures
Author Response
We thank the referee and have added the scale bars in figures 2, 4, and 5 accordingly.
Reviewer 2 Report
General comments:
The authors of this review should be lauded for gathering current technologies in vascular engineering. However, several major changes and the addition of critical references are needed to make the review more organized and comprehensive.
Also, text can be shortened significantly without the loss of important information by judicious editing. As the title implies that this review highlights engineered vasculature models in the field of cancer and regenerative medicine, the first half part (Part 1; Introduction) of the manuscript could be lessened, and the other half (part 2-3) could be reinforced.
Introducing 3D bioprinting-based methods to generate in vitro vasculature is the novelty of this review differentiated from other reviews mainly focusing on microfluidic-based methods. Since the authors are from a group with expertise in vessel bioprinting, a profound description of this part is expected to be helpful to the community of vascular engineering researchers. I highly recommend emphasizing the part of 1.2.2. and considering changing the section title to reveal bioprinting better.
Specific comments:
- In line 103, Matrigel is mentioned as significant material for modeling in vitro angiogenesis. However, the majority of models recapitulating 3D angiogenesis are based on fibrin or collagen. This sentence can give misconceptions to the readers.
- In lines 106-110, the proangiogenic factors (VEGF and other various angiocrines) or the source of those factors, such as co-cultured fibroblasts, should be considered as another major component in engineering in vitro vasculature.
- In the last paragraph of section 1.1, emerging single-cell RNA-sequencing technology could be included as one of the ways of understanding EC heterogeneity. ("An integrated gene expression landscape profiling approach to identify lung tumor endothelial cell heterogeneity and angiogenic candidates." Cancer cell 37.1 (2020): 21-36.)
- In section 1.2.3., other critical reviews comprehensively covering microfluidic-based models of vascular tissue could be introduced:
"Engineering the multiscale complexity of vascular networks." Nature Reviews Materials 7.9 (2022): 702-716.
"Microfluidic-based vascularized microphysiological systems." Lab on a Chip 18.18 (2018): 2686-2709.
"Microfluidic platforms for mechanobiology." Lab on a Chip 13.12 (2013): 2252-2267. (Especially for section 1.2.3.2.) - In addition to lines 269-270, it will be good to discuss/compare the bioprinting-based method mentioned in ‘1.2.2. Manufacture of vasculature tubes’ (CMST and AMST) with the microfluidic-based method to generate lumen structure. What would be advantageous/disadvantageous for adopting microfluidic technology to generate these structures?
- In addition to the shear flow in vascular tissue mentioned in section 1.2.3.2., interstitial flow also plays a critical role in the homeostasis of the tissue and angiogenesis. This could be discussed by introducing the first microfluidic-based angiogenesis model applying controllable interstitial flow: "Interstitial flow regulates the angiogenic response and phenotype of endothelial cells in a 3D culture model." Lab on a Chip 16.21 (2016): 4189-4199.
- Lines 373-385 could be shortened or replaced with a more tumor-specific introduction. Most of the contents in this part overlap with those mentioned in the front part of the manuscript. A general description of tumor vascular biology or tumor microenvironment could be summarized from these reviews:
"The tumor microenvironment and its role in promoting tumor growth." Oncogene 27.45 (2008): 5904-5912.
"Normalizing function of tumor vessels: progress, opportunities, and challenges." Annual review of physiology 81 (2019): 505-534.
Please also introduce critical reviews covering current MPS techniques for cancer therapeutics. Such as "Normalizing function of tumor vessels: progress, opportunities, and challenges." Annual review of physiology 81 (2019): 505-534. - Title of section 2.2. Drug Toxicity in line 436 should be revised with either ‘drug testing’ or ‘drug efficacy and toxicity evaluation’. Drug efficacy, which is mainly covered in this section, is conceptually different from drug toxicity.
- What is written in lines 443-445 is more relevant to section 2. 1..
- As mentioned in the introduction and line 463-464, I highly recommend adding a section or a paragraph introducing current efforts to engineer in vitro lymphatic vasculature. Since the importance of lymphatic vasculature is emerging in tumor biology and therapeutics ("Tumour lymph vessels boost immunotherapy." (2017): 340-342.), the need for in vitro system is in high demand but not matured as blood vessel modeling. Recent in vitro microfluidic-based models in the lymphatic vessel focus on both roles in cancer metastasis and immune surveillance.
"Human tumor-lymphatic microfluidic model reveals differential conditioning of lymphatic vessels by breast cancer cells." Advanced healthcare materials 9.3 (2020): 1900925.
"Lymphatic Vessel Networks: Modeling 3D Human Tumor Lymphatic Vessel Network Using High-Throughput Platform ." Advanced Biology 5.2 (2021): 2170021. - I agree with what is mentioned in line 501. It may be great to discuss this point more to allow readers to rethink balancing between complexity and simplicity when reconstructing in vitro tumor immune microenvironment. That is indeed almost impossible to fully reconstruct in vivo-like immune system. The system could be simplified to dissect specific mechanisms in tumor invasion and interactions of immune cells. One of the emerging studies is: "Macrophages-triggered sequential remodeling of endothelium-interstitial matrix to form pre-metastatic niche in microfluidic tumor microenvironment." Advanced Science 6.11 (2019): 1900195.
- In section 3.1. two critical works from the groups that pioneered establishing perusable 3D in vitro vasculature needs to be included.
"Engineering of functional, perfusable 3D microvascular networks on a chip." Lab on a Chip 13.8 (2013): 1489-1500. à This work is a milestone for making 3D lumenized pefusable vessel using microfluidic device.
"3D microtumors in vitro supported by perfused vascular networks." Scientific reports 6.1 (2016): 31589. à This work introduces a novel method of laminin coating to robustly generate perfusable vasculature. - In the section 3, more references with 3D vasculature or vasculogenesis-promoting in vitro scaffold which contributed to in vivo tissue regeneration should be introduced. Most of the descriptions of each study in section 3.1. and 3.2. focus on how the perfusability is achieved during the vascular engineering. Practical applications on tissue regeneration should be suggested. Recommended review paper and examples of critical studies in this topic are suggested below.
"Strategies to Promote Vascularization in 3D Printed Tissue Scaffolds: Trends and Challenges." Biomacromolecules 23.7 (2022): 2730-2751.
"3D printed complex tissue construct using stem cell-laden decellularized extracellular matrix bioinks for cardiac repair." Biomaterials 112 (2017): 264-274. ïƒ Cardiac tissue regeneration
"3D microphysiological system-inspired scalable vascularized tissue constructs for regenerative medicine." Advanced Functional Materials 32.1 (2022): 2105475. ïƒ Regeneration after hind limb ischemia
"Delivering proangiogenic factors from 3D-printed polycaprolactone scaffolds for vascularized bone regeneration." Advanced healthcare materials 9.23 (2020): 2000727. ïƒ Vascularized bone regeneration
"Human bone marrow-derived mesenchymal stem cells play a role as a vascular pericyte in the reconstruction of human BBB on the angiogenesis microfluidic chip." Biomaterials 279 (2021): 121210. ïƒ Regeneration after stroke
Minor comments:
- The style of references should be unified as recommended by the journal. The current version is highly disorganized.
- In line 100, the reference should be written as [22-23] or [22, 23].
- In line 599, it should be ‘patient-specific’.
- In line 621, “ex vivo” should be written in italic. Also, in vitro is written in either in vitro or in-vitro. Please unify.
- In line 639, it should be ‘vasculogenesis’.
Round 2
Reviewer 2 Report
The authors addressed most of the comments raised by the reviewer, and the manuscript was improved with better organization and adequate references.